# OpenReview forum: "Towards Scalable Oversight with Collaborative Multi-Agent Debate in Error Detection"
_ICLR.cc/2026/Conference — Submitted to ICLR 2026_

### Official Review · Reviewer_Yicp · 2025-10-31

**Soundness:** 3
**Presentation:** 3
**Contribution:** 2
**Rating:** 4
**Confidence:** 4

**Summary:**

This paper investigated with cooperative multi-agent debate can improve error detection in LLM. The authors propose a modified debate process and provide some theoretical results related to their method.

**Strengths:**

- **[Motivation]** The issue of "debate hacking" or "judge hacking" is well known issue when using external judging mechanisms (like an LLM) rather than ground truth.
- **[Empirical Coverage]** The authors perform experiments on a multitude of LLMs covering different families and levels of strength.
- **[Clarity]** Writing is clear and figures (especially the methodology figures) effectively communicate the core ideas.
- **[Improvements over prior works]** The authors method typically outperforms baselines (albeit to a fairly small degree in most cases).

**Weaknesses:**

- **[Incremental novelty]** The main contribution (collaboration instead of competition) is conceptually minor and is achieved by a simple change in prompting style and framing, not a new algorithmic mechanism.

- **[Trivial and uninsightful theory]** Theoretical results (Props. 2.2–2.3) are essentially tautologies: cooperation lowers Bayes risk if it adds mutual information. Results don't need to "complex" if they provide useful insight, but these results are both vacuous and uninformative.

- **[Conceptual overlap]** The method closely resembles prior cooperative or consensus-based debate frameworks such as **Society of Mind (Du et al., 2023)** and merly adapts these the setting of oversight with little meaningful change.

- **[Results]** While the authors provide comprehensive results,  most improvements are quite small (1–4%). No confidence margins are given, which makes it difficult to understand if these minor improvements are even statistically significant.

- **[Limited insight and ablations]** 1) No ablation on which prompt component drives improvements.

- **[“human alignment” analysis]** When discussing Fig. 5b, the authors states "ColMAD yields more human-aligned explanations and reasoning", yet the results in this figure use only LLM-as-a-judge evaluation.  This claim greatly exaggerates the authors' method.

- **[Lack of cost analysis]** One of the major drawbacks of debate (cooperative or competitive) is the added test-time compute. The authors method adds further test-time compute as they introduce additional tasks for the models. The lack of analysis of computational or test-time costs makes it difficult to fully appreciate the authors' method.

**Questions:**

See weaknesses.

---

> ### Author Response · Authors · 2025-11-24
> **Response to Reviewer Yicp (part 1)**
>
> Thank you for your suggestions and time in reviewing our paper. Please find our detailed responses below to your concerns.
>
> > W1&W2&W3 Novelty, theoretical results, and relations with previous cooperative/consensus-based debate:
>
> **A1** We need to clarify that **ColMAD differs fundamentally from previous MAD methods**. First, previous MAD approaches can be categorized as competitive MAD (CopMAD) [1,2,3,4] and consensus-seeking MAD (CosMAD) [5,6] (A full table of the related work can be found in [the response to Reviewer 3NPC](https://openreview.net/forum?id=W6qSjvTQMW&noteId=Vmg3OmjWGp)).
>
> Given the categorization, we show that **ColMAD has provable theoretical advantages over CopMAD and CosMAD**:
> - Basic MAD setting: Given a question $Q$ with a binary label $Y\in\{0,1\}$, two debaters $A$ and $B$ with different beliefs of the label that debate with each other and produce messages $M=(m_A,m_B)$. Without loss of generality, $A$ believes the answer $Y=1$ with initial rationale $x_A$ and $B$ believes $Y=0$ with initial rationale $x_B$. A judge $J$ will give Bayes-optimal predictions $Y_J=J(X_0,M)$ based on the information $X_0$ and $M$ (Assumption 2.1). When we consider the number of debating rounds, we will denote the messages generated before round $t$ as $M^{(t-1)}$; otherwise, we will omit it for the simplicity of notations.
>
> - **CopMAD**: Debaters $A$ and $B$ have opposed utilities that convince the judge $J$ to their respective answers, i.e., $$u_A(m_A)=I(Y_J=Y_A;m_A|X_0,M^{(t-1)}),  u_B(m_B)=P(Y_J=Y_A;m_B|X_0,M^{(t-1)}).$$
> - *Equilibrium of CopMAD*: CopMAD can be considered as a **natural extension of the typical cheap talk game in game theory[7]**, where debaters $A$ and $B$ can transmit costless messages to each other and to the judge $J$. Then there exists a babbling equilibrium:
>    - At each round $t$, given the historical messages $(X_0,M^{(t-1)}$, each debater’s optimal strategy is to diminish the effect of $(X_0,M^{(t-1)}$ to the judge regardless the facts. At babbling equilibrium, both debaters send state-independent messages and the transcript carries no information about $Y$, i.e., $I(Y;M|X_0)=0$. It also explains the existence of **debate hacking, as there is no cost of babbling**.
>   - Given the circumstance, for the judge, the optimal strategy is to ignore all the information of the debating, and make the prediction based on the initial responses, which gives the risk as $R(X_0)$.
> - **ColMAD**: Debaters $A$ and $B$ have the same utilities that $u(A)=u(B)=I(Y;m_A|X_0,M^{(t-1)})$, i.e., complementing missing messages and reducing uncertainty about the truth $Y$.
> - *Equilibrium of ColMAD*: Given the utility functions of debaters, the equilibrium of ColMAD satisfies $I(Y;M|X_0)\geq 0$, which gives no worse risk than CopMAD.
> - **CosMAD**: The utilities of debaters $A$ and $B$ are the same as ColMAD, however, the consensus-seeking procedure will only consider messages under consensus. Hence, the judge will only see a subset of the debating transcript to make predictions $Y_J = J(\Phi_{\cap}(X_0,M)$, where $\Phi_{\cap}$ is a filter function that filters out information $A$ and $B$ did not reach a consensus.
> - *Equilibrium of CosMAD*: Using the information processing inequality, we have $I(Y;\Phi_{\cap}(M)|X_0)\leq I(Y;M|X_0)$. Therefore, we know that ColMAD is no worse than CosMAD.

---

> ### Author Response · Authors · 2025-11-24
> **Response to Reviewer Yicp (part 2)**
>
> We have also extended the experiments to include more MAD baselines, including SoM[5] and MP[4]. The results are given as follows, where we can find that **ColMAD still largely outperforms the previous MAD methods, including SoM and MP, aligned to the theoretical advantages we show before**:
>
> |                 |                 |          | math  |       | finegrained |       | answerability |       | Avg. F1 | Avg. F2 |
> |-----------------|-----------------|----------|-------|-------|-------------|-------|---------------|-------|---------|---------|
> | Debater#1       | Debater#2       | protocol | F1    | F2    | F1          | F2    | F1            | F2    |         |         |
> |                 |                 | CopMAD   | 66.67 | 65.97 |       66.24 | 63.11 |         50.37 | 42.93 |   61.09 |   57.34 |
> |                 |                 | SoM      | 80.77 | 89.55 |       75.00 | 78.59 |         60.87 | 58.06 |   72.21 |   75.40 |
> |                 |                 | MP       |    76 | 82.43 |       73.56 | 74.59 |         52.78 | 46.91 |   67.45 |   67.98 |
> |                 |                 | Ensemble | 78.34 | 88.91 |       75.62 | 83.33 |         68.78 | 72.22 |   74.25 |   81.49 |
> | Llama3.1-70B    | GPT4o-mini      | ColMAD   | 78.38 | 90.06 |       77.42 | 88.98 |         72.91 | 79.74 |   76.24 |   86.26 |
> |                 |                 | CopMAD   | 78.34 | 88.91 |       73.79 | 82.43 |         68.16 | 69.32 |   73.43 |   80.22 |
> |                 |                 | SoM      | 80.77 | 89.55 |       75.27 | 79.37 |         62.58 | 60.14 |   72.87 |   76.35 |
> |                 |                 | MP       | 74.03 | 75.79 |       71.35 | 75.00 |         60.13 | 55.56 |   68.50 |   68.78 |
> |                 |                 | Ensemble | 78.34 | 88.91 |       75.62 | 83.33 |         68.78 | 72.22 |   74.25 |   81.49 |
> | GPT4o-mini      | Mistral-7B-v0.3 | ColMAD   | 77.13 | 88.84 |       77.14 | 87.10 |         74.40 | 82.26 |   76.22 |   86.07 |
> |                 |                 | CopMAD   | 74.73 | 76.75 |       59.35 | 56.10 |         55.03 | 50.00 |   63.04 |   60.95 |
> |                 |                 | SoM      | 59.35 | 55.29 |       73.02 | 77.70 |         54.67 | 49.88 |   62.35 |   60.96 |
> |                 |                 | MP       | 78.05 | 85.84 |       72.83 | 76.31 |         53.16 | 50.12 |   68.01 |   70.76 |
> |                 |                 | Ensemble | 71.20 | 75.22 |       74.04 | 83.15 |         64.04 | 64.92 |   69.76 |   74.43 |
> | Llama3.1-70B    | Mistral-7B-v0.3 | ColMAD   | 77.83 | 89.21 |       75.58 | 86.86 |         70.53 | 74.28 |   74.65 |   83.45 |
> |                 |                 | CopMAD   | 74.64 | 82.98 |       70.53 | 75.28 |         64.37 | 64.37 |   69.85 |   74.21 |
> |                 |                 | SoM      | 57.86 | 54.76 |       75.00 | 80.54 |         56.58 | 52.06 |   63.15 |   62.45 |
> |                 |                 | MP       | 73.02 | 76.67 |       70.33 | 73.23 |         57.72 | 52.44 |   67.02 |   67.45 |
> |                 |                 | Ensemble | 69.11 | 73.01 |       73.93 | 83.69 |         62.50 | 62.93 |   68.51 |   73.21 |
> | GPT4o-mini      | DeepSeek-R1     | ColMAD   | 82.76 | 90.52 |       76.77 | 83.89 |         70.79 | 71.75 |   76.77 |   82.05 |
> |                 |                 | CopMAD   | 49.59 | 39.27 |       28.57 | 21.80 |         18.69 | 13.59 |   32.28 |   24.89 |
> |                 |                 | SoM      | 81.72 | 85.01 |       76.92 | 76.65 |         56.95 | 52.18 |   71.86 |   71.28 |
> |                 |                 | MP       | 72.41 | 72.41 |       51.95 | 48.90 |         34.15 | 27.34 |   52.84 |   49.55 |
> |                 |                 | Ensemble | 81.22 | 87.34 |       79.57 | 83.90 |         65.91 | 66.36 |   75.57 |   79.20 |
> | Llama3.1-70B    | DeepSeek-R1     | ColMAD   | 81.95 | 90.13 |       78.26 | 87.66 |         73.91 | 76.40 |   78.04 |   84.73 |
> |                 |                 | CopMAD   | 52.86 | 46.13 |       65.96 | 69.98 |         60.81 | 55.01 |   59.88 |   57.04 |
> |                 |                 | SoM      | 85.71 | 86.01 |       76.47 | 76.47 |         59.31 | 52.96 |   73.83 |   71.81 |
> |                 |                 | MP       | 31.48 | 23.04 |       47.15 | 38.36 |         42.52 | 34.79 |   40.38 |   32.06 |
> |                 |                 | Ensemble | 80.81 | 87.15 |       80.85 | 85.78 |         65.48 | 64.10 |   75.71 |   79.01 |

---

> ### Author Response · Authors · 2025-11-24
> **Response to Reviewer Yicp (part 3)**
>
> > W3 Confidence margins
>
> **A2** To facilitate reproducibility, **we set the temperature to 0 for all LLMs in our experiments, which minimizes the randomness**.
> To investigate other setting, we also conducted experiments with GPT4o-mini and Llama3.1-70B with the temperature set to 1 using the seed from $1$ to $10$ to conduct $10$ different runs. We first report the mean and standard deviation in the table below:
>
> |              | math  |      | finegrained |      | answerability |      | avg. F2 |      |
> |--------------|-------|------|-------------|------|---------------|------|---------|------|
> | GPT4o-mini   | 88.82 | 1.18 |       78.71 | 2.11 |         75.06 | 3.67 |   80.86 | 1.54 |
> | Llama3.1-70B | 87.11 | 1.29 |       82.31 | 1.64 |         62.52 | 3.91 |   77.31 | 1.36 |
> | ColMAD       | 89.53 | 0.80 |       85.40 | 1.34 |         75.20 | 3.81 |   83.38 | 1.11 |
> | CopMAD       | 77.00 | 3.53 |       63.77 | 3.84 |         46.14 | 3.93 |   62.30 | 2.01 |
> | Ensemble     | 88.60 | 1.13 |       81.15 | 1.36 |         68.40 | 3.58 |   79.38 | 1.33 |
>
> Furthermore, we obtained 95% confidence interval as follows:
> | 95% interval | math  |       | finegrained |       | answerability |       | avg. F2 |       |
> |--------------|-------|-------|-------------|-------|---------------|-------|---------|-------|
> | GPT4o-mini   | 87.93 | 89.71 |       77.12 | 80.30 |         72.29 | 77.83 |   79.71 | 82.02 |
> | Llama3.1-70B | 86.14 | 88.08 |       81.07 | 83.54 |         59.57 | 65.46 |   76.29 | 78.33 |
> | ColMAD       | 88.93 | 90.13 |       84.39 | 86.41 |         72.33 | 78.07 |   82.54 | 84.21 |
> | CopMAD       | 74.34 | 79.65 |       60.87 | 66.67 |         43.17 | 49.10 |   60.78 | 63.82 |
> | Ensemble     | 87.74 | 89.45 |       80.12 | 82.17 |         65.70 | 71.10 |   78.38 | 80.38 |
>
> It can be found that **ColMAD is the only method that achieves a non-overlapped 95% confidence interval over the single-agent method**. In the task of math word problem generation, there is a slight overlap between ColMAD and Ensemble, for which we obtained a p-value of 6%. It is because both GPT4o-mini and Llama3.1-70B are relatively good at the task; therefore, the performance is relatively saturated.
>
> > W4 Ablation on prompt components.
>
> **A3** We ablate three sub-parts of our prompt design, including the quote-system, confidence declaration, and self-auditing with GPT4o-mini and Llama3.1-70B. The results are given below, where we can find that each part of the prompt helps facilitate the error detection:
> |                                                           | math  |       | finegrained |       | answerability |       | Avg. F1 | Avg. F2 |
> |-----------------------------------------------------------|-------|-------|-------------|-------|---------------|-------|---------|---------|
> | protocol                                                  | F1    | F2    | F1          | F2    | F1            | F2    |         |         |
> | ColMAD                                                    | 78.38 | 90.06 |       77.42 | 88.98 |         72.91 | 79.74 |   76.24 |   86.26 |
> | w/o quote-system                                          | 77.68 | 89.69 |       77.21 | 88.30 |         73.17 | 80.47 |   76.02 |   86.15 |
> | w/o quote-system & confidence declaration                 | 77.83 | 89.21 |       76.42 | 86.72 |         74.04 | 82.09 |   76.10 |   86.01 |
> | w/o quote-system & confidence declaration & self-auditing | 78.03 | 89.88 |       74.64 | 84.05 |         75.49 |  82.8 |   76.05 |   85.58 |

---

> ### Author Response · Authors · 2025-11-24
> **Response to Reviewer Yicp (part 4)**
>
> > W5 Human aligned analysis
>
> **A4** We need to clarify that, although we use LLM-as-judge to obtain the results, we use LLMs to **compare the reference human explanations provided in the benchmark[8]** and the responses from different MAD protocols. The LLM merely examines whether the reasoning of the corresponding MAD protocol follows the rationale given by humans.
>
> To provide more evidence, here we analyze the debating results from CopMAD and ColMAD (Llama3.1-70B vs DeepSeek-R1) in math problem generation. We extract some representative mistakes given by human explanations, including `calculation error`, `unnecessary steps`, `specific phase requirement`, `specific integer format`, and count whether they are included in the final reasoning given by  CopMAD and ColMAD, respectively. The results are shown below:
> |                            | Llama3.1-70B | DeepSeek-R1 | CopMAD | ColMAD |
> |----------------------------|--------------|-------------|--------|--------|
> | specific integer format    |        17.60 |       38.20 |  38.20 |  44.10 |
> | specific phase requirement |        50.00 |       58.30 |  58.30 |  58.30 |
> | ambiguous_or_unanswerable  |        30.00 |       40.00 |  30.00 |  40.00 |
> | missing_required_content   |        14.29 |       14.29 |   0.00 |  28.57 |
> | calculation error          |         0.00 |       50.00 |   0.00 |  50.00 |
>
> It can be found that, **ColMAD consistently identifies the same mistakes pointed out by humans better than other methods**.
> In addition, during the development of this project, interestingly, ColMAD can even find mistakes in human explanations. Here is an example in math problem generation:
> First, here is the requirement for generating a math word problem:
> ```
> Generate a math word problem that satisfies the following requirements. You only need to generate a question. You do not need to generate a solution for the question.
> General Requirements:
> The generated question should be valid and answerable.
> The generated question should not be ambiguous and the final answer to the generated questions must be unique.
> The generated question should not include information that is not necessary to answer the question.
> Specific Requirements:
> Generated question should start with: A shrewd opportunist bought 50 equally priced Kale Kids in November for t dollars total.
> The problem requires an understanding of profit calculation in terms of percentage.
> ```
> Here is the response from GPT4
> ```
> A shrewd opportunist bought 50 equally priced Kale Kids in November for t dollars total. In December, he sold all the Kale Kids for a total of 1.5t dollars. What was his profit percentage?
> ```
> **The human labelling in the benchmark considers that the response contains no error. However, with ColMAD, we find that the response indeed contains an error**:
> ```
> 'In December', which is not needed to determine the profit percentage. While the month 'November' is mandated by the required opening sentence, adding 'In December' is extra information that is not necessary to answer the question, violating the general requirement to avoid unnecessary information.
> ```

---

> ### Author Response · Authors · 2025-11-24
> **Response to Reviewer Yicp (part 5)**
>
> > W6 Cost analysis
>
> **A5** We count the averaged token costs if the debating transcripts of different MAD methods for Llama3.1-70B vs GPT4o-mini, given in the table below:
>
> |          | math  |                  | finegrained |                  | answerability |                  |
> |----------|-------|------------------|-------------|------------------|---------------|------------------|
> | protocol | F2    | token cost       | F2          | token cost       | F2            | token cost       |
> | ColMAD   | 90.06 | 13259.15+-971.28 |       88.98 |   13327+-1033.25 |         79.74 | 13323+-1086.06   |
> | CopMAD   | 88.91 | 11449.09+-524.24 |       82.43 | 11484.94+-538.15 |         69.32 | 11401.81+-714.19 |
> | SoM      | 89.55 | 9461.82+-4225.49 |       79.37 | 6642.09+-1061.34 |         60.14 | 7076.89+-4165.38 |
> | MP       | 75.79 | 9690.29+-2224.01 |       75.00 | 9344.22+-712.82  |         55.56 | 9471.56+-2136.85 |
>
> Although ColMAD costs more additional tokens than other methods, the improvements in performance are also significant. Nevertheless, given the high token costs of MAD methods, it will be a promising future direction to investigate more efficient communication methods, such as[9], to avoid the token costs.
>
> **References**
>
> [1] AI safety via debate, arXiv'18.
>
> [2] Debating with more persuasive LLMs leads to more truthful answers, ICML'24.
>
> [3] On scalable oversight with weak LLMs judging strong LLMs, NeurIPS'24.
>
> [4] Encouraging divergent thinking in large language models through multi-agent debate, EMNLP'24.
>
> [5] Improving factuality and reasoning in language models through multi-agent debate, ICML'24.
>
> [6] Reconcile: Round-table conference improves reasoning via consensus among diverse llms, ACL'24.
>
> [7] Strategic Information Transmission, Econometrica, 1982.
>
> [8] Evaluating llms at detecting errors in LLM responses, COLM’24.
>
> [9] Thought Communication in Multiagent Collaboration, NeurIPS’25.

---

> ### Author Response · Authors · 2025-11-27
> **A gentle reminder**
>
> Dear Reviewer Yicp,
>
> Thank you once again for your time and effort in reviewing our work. As the discussion period is only one week away, to allow us sufficient time to address your concerns, could you kindly review our responses and let us know if you still have any questions? Thank you!
>
> Best regards,
> Authors

---

### Official Review · Reviewer_3NPC · 2025-10-31

**Soundness:** 2
**Presentation:** 2
**Contribution:** 2
**Rating:** 2
**Confidence:** 3

**Summary:**

This paper tackles scalable oversight by reframing multi-agent debate for LLM error detection from a zero-sum contest (which the authors show is prone to “debate hacking”) into a collaborative, non-zero-sum protocol called ColMAD. Theoretically, they prove that when agents share information that increases the judge’s mutual information about the label, collaboration weakly dominates competitive debate; conversely, with dishonest agents, competitive MAD degenerates to no gain over the initial rationales. ColMAD adds three prompt-level safeguards—quote-verifiable evidence, self-auditing of potential failure modes, and confidence calibration—to make critiques more informative and less manipulative. On the ReaLMistake benchmark, ColMAD consistently improves F1/F2 over strong single-agent baselines and outperforms competitive MAD by large margins.

**Strengths:**

* This paper applies MAD in error detection, which extends the application boundary of MAD systems
* The evaluation is comprehensive covering a range of LLMs and benchmarks, demonstrating the superior improvement
* The paper is well written and easy to follow

**Weaknesses:**

* The argument "as previous approaches often frame MAD as a zero-sum game where the debaters compete with each other" is not convincing. I believe most MAD systems are not framed as zeros-sum games. There lacks references or empirical evidences to support this argument. While a part of MAD systems encourage agents to debate against each other, they cannot be considered strictly as zero-sum game as well.
* The major contribution, "ColMAD asks debaters to collaborate and complement each other’s missing points" is not something novel in MAD design. Actually, most MAD systems do not explicitly encourage debators to play against each other.
* In the evaluation, only CopMAD, ColMAD, and Ensemble are considered. As discussed in previous weaknesses, a lot of previous MAD methods are not considered. To name a few, [1][2][3].
* Based on these points, ColMAD is approximately a direct application of MAD methods in error detection. The challenge (W.1) is not convincing, and the method (W.2) does not obviously dispart from previous practice.

[1]Exchange-of-Thought: Enhancing Large Language Model Capabilities through Cross-Model Communication
[2]Encouraging Divergent Thinking in Large Language Models through Multi-Agent Debate
[3]ReConcile: Round-table conference improves reasoning via consensus among diverse LLMs

**Questions:**

N/A

---

> ### Author Response · Authors · 2025-11-24
> **Response to Reviewer 3NPC (part 1)**
>
> Thank you for your time and valuable comments about our paper. Please find our responses below to your concerns.
>
> > W1 Zero-sum formulation of the existing MAD methods.
>
> **A1** Regarding the zero-sum formulation, we refer to the line of MAD works that show the potential of MAD for scalable oversight, such as [1,2,3,4] and so on. In addition to the CopMAD, the line of the other MAD works can be considered as consensus-seeking MAD (CosMAD) [5,6].
>
> The referred works [4] (corresponding to [2] in the review) can be considered as CopMAD with assigned personas. [6] (corresponding to [1] in the review) can be considered as CSD, which extends SoM to more agents, where the extension is orthogonal to our work, as we focus on a two-debater agent setting. [7] (corresponding to [3] in the review) focuses on improving the communication of multiple agents, which is orthogonal to our work. We have revised our manuscript to include the following table of existing work to make the position of our work clearer:
> | Name                     | MAD aspect                            | Paper                                                                                                         |
> |--------------------------|---------------------------------------|---------------------------------------------------------------------------------------------------------------|
> | ColMAD                   | Collaborative MAD                     | Ours                                                                                                          |
> | CopMAD                   | Competitive MAD                       | On scalable oversight with weak LLMs judging strong LLMs, NeurIPS'24                                          |
> | Khan et al., 2024        | Competitive MAD                       | Debating with more persuasive LLMs leads to more truthful answers, ICML'24                                    |
> | Irving et al., 2018      | Competitive MAD                       | AI safety via debate, arXiv'18                                                                                |
> | Brown-Cohen et al., 2025 | Competitive MAD                       | Avoiding Obfuscation with Prover-Estimator Debate, arXiv'25                                                   |
> | Buhl et al., 2025        | Competitive MAD                       | An alignment safety case sketch based on debate, arXiv'25                                                     |
> | Brown-Cohen et al., 2024 | Competitive MAD                       | Scalable AI safety via doubly-efficient debate, ICML'24                                                       |
> | Rahman et al., 2025      | Competitive MAD                       | AI Debate Aids Assessment of Controversial Claims, NeurIPS'25                                                 |
> | Carro et al., 2025       | Competitive MAD                       | AI Debaters are More Persuasive when Arguing in Alignment with Their Own Beliefs, arXiv'25                    |
> | MP                       | Competitive MAD with persona          | Encouraging divergent thinking in large language models through multi-agent debate, EMNLP'24                  |
> | SoM                      | Consensus-seeking MAD                 | Improving factuality and reasoning in language models through multiagent debate, ICML'24                      |
> | Reconcile                | Consensus-seeking MAD                 | Reconcile: Round-table conference improves reasoning via consensus among diverse llms, ACL'24                 |
> | CoMM                     | Consensus-seeking MAD with persona    | CoMM: Collaborative multi-agent, multi-reasoning-path prompting for complex problem solving, NAACL'24.        |
> | EoT                      | Communication of MAD                  | Exchange-of-Thought: Enhancing Large Language Model Capabilities through Cross-Model Communication, EMNLP'23  |
> | ChatEval                 | Application of MAD                    | ChatEval: Towards Better LLM-based Evaluators through Multi-Agent Debate, ICLR'24                             |
> | Wang et al., 2024        | Benchmarking of MAD                   | Rethinking the bounds of LLM reasoning: Are multi-agent discussions the key? ACL'24                           |
> | Smit et al., 2024        | Benchmarking of MAD                   | Should we be going mad? a look at multi-agent debate strategies for llms, ICML'24                             |
> | Wynn et al., 2025        | Benchmarking of Consensus-seeking MAD | Talk Isn’t Always Cheap: Understanding Failure Modes in Multi-Agent Debate, ICML'25 Workshop                  |
> | Zhang et al., 2025       | Benchmarking of MAD                   | If Multi-Agent Debate is the Answer, What is the Question? arXiv'25                                           |
> | Yang et al., 2025        | Benchmarking of MAD                   | Revisiting Multi-Agent Debate as Test-Time Scaling: A Systematic Study of Conditional Effectiveness, arXiv'25 |

---

> ### Author Response · Authors · 2025-11-24
> **Response to Reviewer 3NPC (part 2)**
>
> > W2 Novelty of ColMAD and relation to existing works.
>
> **A2** As in **A1**, existing works can be categorized as CopMAD and CosMAD, where the former encourages agents to compete with each other, and the latter encourages agents to seek consensus through a discussion. Theoretically, one could formulate CopMAD, ColMAD, and CosMAD as follows:
> - Basic MAD setting: Given a question $Q$ with a binary label $Y\in\{0,1\}$, two debaters $A$ and $B$ with different beliefs of the label that debate with each other and produce messages $M=(m_A,m_B)$. Without loss of generality, $A$ believes the answer $Y=1$ with initial rationale $x_A$ and $B$ believes $Y=0$ with initial rationale $x_B$. A judge $J$ will give Bayes-optimal predictions $Y_J=J(X_0,M)$ based on the information $X_0$ and $M$ (Assumption 2.1). When we consider the number of debating rounds, we will denote the messages generated before round $t$ as $M^{(t-1)}$; otherwise, we will omit it for the simplicity of notations.
>
> - **CopMAD**: Debaters $A$ and $B$ have opposed utilities that convince the judge $J$ to their respective answers, i.e., $$u_A(m_A)=I(Y_J=Y_A;m_A|X_0,M^{(t-1)}),  u_B(m_B)=P(Y_J=Y_A;m_B|X_0,M^{(t-1)}).$$
> - *Equilibrium of CopMAD*: CopMAD can be considered as a **natural extension of the typical cheap talk game in game theory[9]**, where debaters $A$ and $B$ can transmit costless messages to each other and to the judge $J$. Then there exists a babbling equilibrium:
>    - At each round $t$, given the historical messages $(X_0,M^{(t-1)}$, each debater’s optimal strategy is to diminish the effect of $(X_0,M^{(t-1)}$ to the judge regardless the facts. At babbling equilibrium, both debaters send state-independent messages and the transcript carries no information about $Y$, i.e., $I(Y;M|X_0)=0$. It also explains the existence of **debate hacking, as there is no cost of babbling**.
>   - Given the circumstance, for the judge, the optimal strategy is to ignore all the information of the debating, and make the prediction based on the initial responses, which gives the risk as $R(X_0)$.
> - **ColMAD**: Debaters $A$ and $B$ have the same utilities that $u(A)=u(B)=I(Y;m_A|X_0,M^{(t-1)})$, i.e., complementing missing messages and reducing uncertainty about the truth $Y$.
> - *Equilibrium of ColMAD*: Given the utility functions of debaters, the equilibrium of ColMAD satisfies $I(Y;M|X_0)\geq 0$, which gives no worse risk than CopMAD.
> - **CosMAD**: The utilities of debaters $A$ and $B$ are the same as ColMAD, however, the consensus-seeking procedure will only consider messages under consensus. Hence, the judge will only see a subset of the debating transcript to make predictions $Y_J = J(\Phi_{\cap}(X_0,M)$, where $\Phi_{\cap}$ is a filter function that filters out information $A$ and $B$ did not reach a consensus.
> - *Equilibrium of CosMAD*: Using the information processing inequality, we have $I(Y;\Phi_{\cap}(M)|X_0)\leq I(Y;M|X_0)$. Therefore, we know that ColMAD is no worse than CosMAD.

---

> ### Author Response · Authors · 2025-11-24
> **Response to Reviewer 3NPC (part 3)**
>
> > W3. More MAD baselines.
>
> **A3** As discussed in **A1**, the referred works ([1,3] in the review) are orthogonal to our work. Hence, we incorporate two additional baselines in the evaluation, SoM[5] and MP[4] ([2] in the review). The results are given as follows, where we can find that **ColMAD still largely outperforms the previous MAD methods, including SoM and MP, aligned to the theoretical advantages we show in A2**:
>
> |                 |                 |          | math  |       | finegrained |       | answerability |       | Avg. F1 | Avg. F2 |
> |-----------------|-----------------|----------|-------|-------|-------------|-------|---------------|-------|---------|---------|
> | Debater#1       | Debater#2       | protocol | F1    | F2    | F1          | F2    | F1            | F2    |         |         |
> | GPT4o-mini      | Llama3.1-70B    | ColMAD   | 78.38 | 90.06 |       75.12 | 85.47 |         75.36 | 83.33 |   76.29 |   86.29 |
> |                 |                 | CopMAD   | 66.67 | 65.97 |       66.24 | 63.11 |         50.37 | 42.93 |   61.09 |   57.34 |
> |                 |                 | SoM      | 80.77 | 89.55 |       75.00 | 78.59 |         60.87 | 58.06 |   72.21 |   75.40 |
> |                 |                 | MP       |    76 | 82.43 |       73.56 | 74.59 |         52.78 | 46.91 |   67.45 |   67.98 |
> |                 |                 | Ensemble | 78.34 | 88.91 |       75.62 | 83.33 |         68.78 | 72.22 |   74.25 |   81.49 |
> | Llama3.1-70B    | GPT4o-mini      | ColMAD   | 78.38 | 90.06 |       77.42 | 88.98 |         72.91 | 79.74 |   76.24 |   86.26 |
> |                 |                 | CopMAD   | 78.34 | 88.91 |       73.79 | 82.43 |         68.16 | 69.32 |   73.43 |   80.22 |
> |                 |                 | SoM      | 80.77 | 89.55 |       75.27 | 79.37 |         62.58 | 60.14 |   72.87 |   76.35 |
> |                 |                 | MP       | 74.03 | 75.79 |       71.35 | 75.00 |         60.13 | 55.56 |   68.50 |   68.78 |
> |                 |                 | Ensemble | 78.34 | 88.91 |       75.62 | 83.33 |         68.78 | 72.22 |   74.25 |   81.49 |
> | GPT4o-mini      | Mistral-7B-v0.3 | ColMAD   | 77.13 | 88.84 |       77.14 | 87.10 |         74.40 | 82.26 |   76.22 |   86.07 |
> |                 |                 | CopMAD   | 74.73 | 76.75 |       59.35 | 56.10 |         55.03 | 50.00 |   63.04 |   60.95 |
> |                 |                 | SoM      | 59.35 | 55.29 |       73.02 | 77.70 |         54.67 | 49.88 |   62.35 |   60.96 |
> |                 |                 | MP       | 78.05 | 85.84 |       72.83 | 76.31 |         53.16 | 50.12 |   68.01 |   70.76 |
> |                 |                 | Ensemble | 71.20 | 75.22 |       74.04 | 83.15 |         64.04 | 64.92 |   69.76 |   74.43 |
> | Llama3.1-70B    | Mistral-7B-v0.3 | ColMAD   | 77.83 | 89.21 |       75.58 | 86.86 |         70.53 | 74.28 |   74.65 |   83.45 |
> |                 |                 | CopMAD   | 74.64 | 82.98 |       70.53 | 75.28 |         64.37 | 64.37 |   69.85 |   74.21 |
> |                 |                 | SoM      | 57.86 | 54.76 |       75.00 | 80.54 |         56.58 | 52.06 |   63.15 |   62.45 |
> |                 |                 | MP       | 73.02 | 76.67 |       70.33 | 73.23 |         57.72 | 52.44 |   67.02 |   67.45 |
> |                 |                 | Ensemble | 69.11 | 73.01 |       73.93 | 83.69 |         62.50 | 62.93 |   68.51 |   73.21 |
> | GPT4o-mini      | DeepSeek-R1     | ColMAD   | 82.76 | 90.52 |       76.77 | 83.89 |         70.79 | 71.75 |   76.77 |   82.05 |
> |                 |                 | CopMAD   | 49.59 | 39.27 |       28.57 | 21.80 |         18.69 | 13.59 |   32.28 |   24.89 |
> |                 |                 | SoM      | 81.72 | 85.01 |       76.92 | 76.65 |         56.95 | 52.18 |   71.86 |   71.28 |
> |                 |                 | MP       | 72.41 | 72.41 |       51.95 | 48.90 |         34.15 | 27.34 |   52.84 |   49.55 |
> |                 |                 | Ensemble | 81.22 | 87.34 |       79.57 | 83.90 |         65.91 | 66.36 |   75.57 |   79.20 |
> | Llama3.1-70B    | DeepSeek-R1     | ColMAD   | 81.95 | 90.13 |       78.26 | 87.66 |         73.91 | 76.40 |   78.04 |   84.73 |
> |                 |                 | CopMAD   | 52.86 | 46.13 |       65.96 | 69.98 |         60.81 | 55.01 |   59.88 |   57.04 |
> |                 |                 | SoM      | 85.71 | 86.01 |       76.47 | 76.47 |         59.31 | 52.96 |   73.83 |   71.81 |
> |                 |                 | MP       | 31.48 | 23.04 |       47.15 | 38.36 |         42.52 | 34.79 |   40.38 |   32.06 |
> |                 |                 | Ensemble | 80.81 | 87.15 |       80.85 | 85.78 |         65.48 | 64.10 |   75.71 |   79.01 |

---

> ### Author Response · Authors · 2025-11-24
> **Response to Reviewer 3NPC (part 4)**
>
> > W4. ColMAD is approximately a direct application of MAD methods in error detection.
>
> **A4** As summarized in **A1** and **A2**, ColMAD differs fundamentally from the existing practice of MAD methods. ColMAD has theoretical advantages over previous CopMAD and CosMAD methods. In addition, the empirical results in **A3** also show the empirical advantages of ColMAD over previous MAD methods.
>
> Please feel free to let us know if you have any other concerns!
>
> **References**
>
> [1] AI safety via debate, arXiv'18.
>
> [2] Debating with more persuasive LLMs leads to more truthful answers, ICML'24.
>
> [3] On scalable oversight with weak LLMs judging strong LLMs, NeurIPS'24.
>
> [4] Encouraging divergent thinking in large language models through multi-agent debate, EMNLP'24.
>
> [5] Improving factuality and reasoning in language models through multi-agent debate, ICML'24.
>
> [6] Reconcile: Round-table conference improves reasoning via consensus among diverse llms, ACL'24.
>
> [7] Exchange-of-Thought: Enhancing Large Language Model Capabilities through Cross-Model Communication, EMNLP'23.
>
> [8] ChatEval: Towards Better LLM-based Evaluators through Multi-Agent Debate, ICLR'24.
>
> [9] Strategic Information Transmission, Econometrica, 1982.

---

> ### Author Response · Authors · 2025-11-27
> **A gentle reminder**
>
> Dear Reviewer 3NPC,
>
> Thank you once again for your time and effort in reviewing our work. As the discussion period is only one week away, to allow us sufficient time to address your concerns, could you kindly review our responses and let us know if you still have any questions? Thank you!
>
> Best regards,
> Authors

---

### Official Review · Reviewer_nyiP · 2025-11-01

**Soundness:** 3
**Presentation:** 3
**Contribution:** 3
**Rating:** 8
**Confidence:** 2

**Summary:**

The paper introduces a new protocol, ColMAD, aimed at improving error detection in large language models (LLMs) through collaborative multi-agent debate. ColMAD reframes multi-agent debate (MAD) as a non-zero-sum game, encouraging agents to support each other rather than compete. Empirical results show ColMAD outperforms previous competitive MAD methods by 19% and improves single-agent methods in error detection. The study highlights the limitations of traditional MAD approaches, which often lead to debate hacking and misleading claims.

**Strengths:**

1. **Clear motivation and observation.**  Competitive MAD protocols often result in performance degradation due to their zero-sum nature.
- Debaters in competitive MAD may misinterpret tasks and present overconfident claims, leading to misleading outcomes.
- Debate hacking behaviors, such as fake evidence and fallacious arguments, are prevalent in competitive settings.

2. **ColMAD.** This paper proposes a new MAD protocol called Collaborative Multi-Agenet Debate (ColMAD) that reframes MAD as a non-zero sum game. ColMAD incorporates specific strategies to enhance the effectiveness of collaborative debates.
- Evidence verification through a quote-based system ensures claims are supported by context.
- Self-auditing requires debaters to identify potential failure modes in their claims.
- Confidence calibration asks debaters to provide confidence estimates for their assertions, improving the robustness of the debate process.

3. **Comprehensive experiments.** ColMAD demonstrates superior error detection capabilities compared to CopMAD, emphasizing the benefits of collaborative approaches. Fig. 2, Fig. 3, Tables 1 & 2 are promising.

**Weaknesses:**

1. From Tables 1 and 2, ColMAD shows substantially better performance than CopMAD but only slightly outperforms the Ensemble baseline. The paper would benefit from a deeper analysis of this comparison. For example, discussing why Ensemble achieves similar results and what unique advantages ColMAD provides beyond simple model aggregation.

**Questions:**

Refer to Weakness.

---

> ### Author Response · Authors · 2025-11-24
> **Response to Reviewer nyiP**
>
> We thank you for your time in reviewing our work and for your positive recommendation of our work. Please find our response to your question below.
>
> > W1 Deeper understanding and unique advantages of ColMAD than Ensemble.
>
> **A1** First of all, following the suggestion by Reviewer qPyM and Yicp, we run the experiments (GPT4o-mini vs Llama3.1-70B) multiple times with the temperature set to 1 to obtain 95% confidence interval (originally it’s set to 0). The results show that, **although ColMAD does not bring much more improvement in terms of numerical values, the improvements in terms of average F2 are significant over the Ensemble and single-agent methods**. In contrast, Ensemble does not bring significant improvements over single-agent methods.
>
> When comparing ColMAD and Ensemble, intuitively, the unique advantages of ColMAD over Ensemble lie in tackling the cases where two debaters have different opinions, and the debating could elicit the necessary information for the judge to make a well-informed decision. The heterogeneity and the capabilities of the debaters play a key role:
> - For cases where ColMAD performs competitively with Ensemble, one can often observe that the agents may have limited heterogeneity. For example, GPT4o-mini vs Llama3.1-70B, if one refers to Figure 2(a), which shows the percent of reduced errors between GPT4o-mini vs Llama3.1-70B is 5%, is much smaller than those in other tasks.
> - As a comparison, Figure 2 also indicates that debating with the Mistral model could bring the largest improvements, which is also reflected in our main results, though Mistral model itself is much weaker than other LLMs.
> - Moreover, given the limited heterogeneity, the upper bound is also limited. In fact, the oracle collaboration of GPT4o-mini vs Llama3.1-70B in math problem generation following Eq. (4) gives the F2 score of 96%, while the simple Ensemble method already reaches a saturated performance.

---

> ### Author Response · Authors · 2025-11-27
> **A gentle reminder**
>
> Dear Reviewer nyiP,
>
> Thank you once again for your time and effort in reviewing our work. As the discussion period is only one week away, to allow us sufficient time to address your concerns, could you kindly review our responses and let us know if you still have any questions? Thank you!
>
> Best regards,
> Authors

---

### Official Review · Reviewer_qPyM · 2025-11-01

**Soundness:** 1
**Presentation:** 1
**Contribution:** 1
**Rating:** 2
**Confidence:** 5

**Summary:**

This paper studies the scalable oversight problem, where the goal is to design methods to supervise powerful AIs using weaker judges. This work continues a line of work that has studied the use of debate between competing AIs to allow for accurate judgements. The submission proposes collaborative rather than competitive debate as an improved scalable oversight method. The authors provide theoretical motivation as well as experimental results with LLMs in  an error-detection task for to demonstrate the advantages of collaborative debate.

**Strengths:**

Exploring new debate protocols empirically and theoretically is an important topic. The paper attempts to formalize situations in which collaborative debate outperforms competitive debate.

**Weaknesses:**

1. The theoretical results in this paper do not really prove anything, and are difficult to parse as the assumptions are not clearly stated.

First, Proposition 1 shows that competitive debate does not improve over no debate at all. The assumption required for this is not stated in the statement of the proposition, but if we read the proof in the appendix, we find that the assumption required is: the competing debaters' equilibrium strategy provides no information to the judge. Proposition 2 shows that collaborative debate does improve over no debate at all. This time the assumption is explicit in the statement: the collaborating debaters' equilibrium strategy provides positive information to the judge. To summarize, if we assume that collaborative debate provides information at equilibrium but competitive debate does not, then of course collaborative debate is better.
Nothing has been proved here. The authors don't even give a definition of competitive or collaborative debate. It is possible that there is some formalization of the collaborative debate game such that the equilibrium there provides more information than the equilibrium in the competitive debate game. Such a result could be interesting, but nothing like that occurs in this paper.

2. The empirical results have significant methodological problems.

The main results in Table 1 are for a setting where debater 1 and the judge are the same model. This is not the right setup for scalable oversight! As mentioned in the summary, the goal of scalable oversight is to help a weaker judge supervise stronger debaters, but in this case the judge is just as strong as the debaters. In the case of an equally strong judge it makes sense that prompting the debaters to try to cooperate to get the right answer could be helpful. This is just a scaffolding that gives more inference-time compute.

Furthermore, it makes no sense to test competitive debate in a setting where the two debater models are different, as is done in Figure 1. The whole point of competitive debate is that two equally powerful models should be able to point out flaws in each others' arguments to help a weaker judge model find the true answer. If the models have different capability levels, this would be expected to significantly harm the performance of competitive debate. In any case, competitive debate is designed for scalable oversight, where the judge is weaker than the debaters, whereas this paper has the judge be as strong as the debater 1, which is not scalable oversight.

None of the results have error-bars for 95% confidence intervals, so it is completely unclear if they are statistically significant. For example, the F1 scores in Figure 3 (a) and (b) for all but the SoM method look like they probably do not significantly differ. Since F1 is in many ways the most important summary metric here, this is a significant issue.

The diagonal of Figure 2 has all zeros on it. Presumably this is because it was not actually tested how two copies of the same model would do in the collaborative debate. However, you could of course run the collaborative debate with two copies and likely get a non-zero error reduction. It's unclear why this was left out, but as mentioned above, this method is just scaffolding that gives more inference time compute, and so two copies of the same model should also perform well. The caption of figure 2 says that models from different companies perform better, but this does not seem like a fair summary of the results. Figure 2 (a) seems to show that Mistral is the best for detecting errors in math, and Figure 2 (b) seems to show that GPT-4 is the best at detecting errors in fact verification. Again, somehow two copies of the same model were not tested against themselves, but I would bet that if you did you would find that the largest entry in (a) would be Mistral-vs-Mistral and in (b) would be GPT-4-vs-GPT-4.

There are many issues with the baseline comparison. Despite mentioning prior empirical work on competitive debate such as that of Khan et al and Kenton et al, this paper does not test their protocol in any of the same tasks. Further, as mentioned above, competitive debate only makes sense with two equal-capability debaters, and a weaker judge. This paper for some reason makes the judge the same as debater 1 and makes debater 2 a different, often weaker model. Further, only the prompts for collaborative debate are shared, and it's completely unclear if all the difference we see is merely due to better prompt optimization for the collaborative setting.

**Questions:**

1. Do you have 95% confidence intervals for any of your results?
2. Why did you set debater 1 = judge? How is this relevant to scalable oversight?
3. Why did you never test your method with two copies of the same model as debaters?
4. Do you have any formalization of the competitive and collaborative debate games where you can actually characterize the equilibria, rather than making an arbitrary assumption about them?

---

> ### Author Response · Authors · 2025-11-24
> **Response to Reviewer qPyM (part 1)**
>
> Thank you for your time and interesting comments! We feel that there is a misunderstanding that would affect your assessment of our work. We would like to collaboratively debate with you to find the truth!
>
> > W1 & Q4 Definitions of the competitive and collaborative debate games and the corresponding equilibria.
>
> **A1** Following the discussion in our manuscript, we now provide definitions of the CopMAD and ColMAD games, and discuss the equilibria :
> To begin with, the basic setting of both games is that, given a question $Q$ with a binary label $Y\in\{0,1\}$, two debaters $A$ and $B$ with different beliefs of the label that debate with each other and produce messages $M=(m_A,m_B)$. Without loss of generality, $A$ believes the answer $Y=1$ with initial rationale $x_A$ and $B$ believes $Y=0$ with initial rationale $x_B$. A judge $J$ will give Bayes-optimal predictions $Y_J=J(X_0,M)$ based on the information $X_0$ and $M$ (Assumption 2.1). When we consider the number of debating rounds, we will denote the messages generated before round $t$ as $M^{(t-1)}$; otherwise, we will omit it for the simplicity of notations.
>
> - CopMAD: Debaters $A$ and $B$ have opposed utilities that convince the judge $J$ to their respective answers, i.e., $$u_A(m_A)=I(Y_J=Y_A;m_A|X_0,M^{(t-1)}),  u_B(m_B)=P(Y_J=Y_A;m_B|X_0,M^{(t-1)}).$$
> - Equilibrium of CopMAD: CopMAD can be considered as a **natural extension of the typical cheap talk game in game theory[1]**, where debaters $A$ and $B$ can transmit costless messages to each other and to the judge $J$. Then there exists a babbling equilibrium:
>    - At each round $t$, given the historical messages $(X_0,M^{(t-1)}$, each debater’s optimal strategy is to diminish the effect of $(X_0,M^{(t-1)}$ to the judge regardless the facts. At babbling equilibrium, both debaters send state-independent messages and the transcript carries no information about $Y$, i.e., $I(Y;M|X_0)=0$. It also explains the existence of **debate hacking, as there is no cost of babbling**.
>   - Given the circumstance, for the judge, the optimal strategy is to ignore all the information of the debating, and make the prediction based on the initial responses, which gives the risk as $R(X_0)$.
> - ColMAD: Debaters $A$ and $B$ have the same utilities that $u(A)=u(B)=I(Y;m_A|X_0,M^{(t-1)})$, i.e., complementing missing messages and reducing uncertainty about the truth $Y$.
> - Equilibrium of ColMAD: Given the utility functions of debaters, the equilibrium of ColMAD satisfies $I(Y;M|X_0)\geq 0$, which gives no worse risk than CopMAD.
>
>
> > W2&Q2 Relation with scalable oversight, which is to help a weaker judge supervise two equally powerful stronger debaters.
>
> **A2** We need to clarify that the central focus of this work is to **reliably detect errors in LLMs’ responses to achieve scalable oversight**,  which follows the spirit of the scalable oversight literature to provide feedback to powerful AI systems[2]. As previous works show that LLMs can not detect the errors themselves[3], and single-agent method also fails short in identifying the errors in LLMs’ responses[4], we resort to multi-agent debate approaches. In order to avoid any potential misunderstandings, we have revised our manuscript to better distinguish our different focus towards scalable oversight from [2].

---

> ### Author Response · Authors · 2025-11-24
> **Response to Reviewer qPyM (part 2)**
>
> > Q3 Why not use two equally powerful debaters? & W3.4 Again, somehow two copies of the same model were not tested against themselves, but I would bet that if you did you would find that the largest entry in (a) would be Mistral-vs-Mistral and in (b) would be GPT-4-vs-GPT-4.
>
> **A3** Regarding the adoption of two equally powerful debaters and a weaker judge, we have conducted the experiments using GPT4o-mini vs GPT4o-mini with GPT4o-mini as the judger, as well as GPT4o-mini vs GPT4o-mini with Mistral as the judger. Now the Ensemble is equivalent to the single-agent performance (i.e., the performance of GPT4o-mini):
> |            |            |            |          | math  |       | finegrained |       | answerability |       | Avg. F1 | Avg. F2 |
> |------------|------------|------------|----------|-------|-------|-------------|-------|---------------|-------|---------|---------|
> | Debater#1  | Debater#2  | Judge      | protocol | F1    | F2    | F1          | F2    | F1            | F2    |         |         |
> | GPT4o-mini | GPT4o-mini | GPT4o-mini | ColMAD   | 76.58 | 87.99 |       72.82 | 78.89 |         70.00 | 75.92 |   73.13 |   80.93 |
> |            |            |            | CopMAD   | 70.90 | 74.44 |       66.28 | 66.74 |         48.23 | 42.29 |   61.80 |   61.16 |
> |            |            |            | Ensemble | 78.70 | 89.10 |       75.76 | 82.78 |         70.10 | 74.73 |   74.85 |   82.20 |
> | GPT4o-mini | GPT4o-mini | Mistral    | ColMAD   | 73.68 | 81.91 |       65.22 | 68.34 |         68.75 | 72.85 |   69.22 |   74.37 |
> |            |            |            | CopMAD   | 61.35 | 58.96 |       54.88 | 53.70 |         53.15 | 47.03 |   56.46 |   53.23 |
> |            |            |            | Ensemble | 78.70 | 89.10 |       75.76 | 82.78 |         70.10 | 74.73 |   74.85 |   82.20 |
>
> One could find that ColMAD still outperforms CopMAD when using the same LLM as the debater, while switching to a weaker judge will reduce the performance. Nevertheless, all the methods underperform the single-agent method (i.e., Ensemble), **demonstrating the necessity of using different LLMs in error detection**.
>
>
> > W3.1 Error bars of 95% confidence interval.
>
> **A4** To facilitate reproducibility, **we set the temperature to 0 for all LLMs in our experiments, which minimizes the randomness**.
>
> To investigate other setting, we also conducted experiments with GPT4o-mini and Llama3.1-70B with the temperature set to 1 using the seed from $1$ to $10$ to conduct $10$ different runs. We first report the mean and standard deviation in the table below:
>
> |              | math  |      | finegrained |      | answerability |      | avg. F2 |      |
> |--------------|-------|------|-------------|------|---------------|------|---------|------|
> | GPT4o-mini   | 88.82 | 1.18 |       78.71 | 2.11 |         75.06 | 3.67 |   80.86 | 1.54 |
> | Llama3.1-70B | 87.11 | 1.29 |       82.31 | 1.64 |         62.52 | 3.91 |   77.31 | 1.36 |
> | ColMAD       | 89.53 | 0.80 |       85.40 | 1.34 |         75.20 | 3.81 |   83.38 | 1.11 |
> | CopMAD       | 77.00 | 3.53 |       63.77 | 3.84 |         46.14 | 3.93 |   62.30 | 2.01 |
> | Ensemble     | 88.60 | 1.13 |       81.15 | 1.36 |         68.40 | 3.58 |   79.38 | 1.33 |
>
> Furthermore, we obtained 95% confidence interval as follows:
> | 95% interval | math  |       | finegrained |       | answerability |       | avg. F2 |       |
> |--------------|-------|-------|-------------|-------|---------------|-------|---------|-------|
> | GPT4o-mini   | 87.93 | 89.71 |       77.12 | 80.30 |         72.29 | 77.83 |   79.71 | 82.02 |
> | Llama3.1-70B | 86.14 | 88.08 |       81.07 | 83.54 |         59.57 | 65.46 |   76.29 | 78.33 |
> | ColMAD       | 88.93 | 90.13 |       84.39 | 86.41 |         72.33 | 78.07 |   82.54 | 84.21 |
> | CopMAD       | 74.34 | 79.65 |       60.87 | 66.67 |         43.17 | 49.10 |   60.78 | 63.82 |
> | Ensemble     | 87.74 | 89.45 |       80.12 | 82.17 |         65.70 | 71.10 |   78.38 | 80.38 |
>
> It can be found that **ColMAD is the only method that achieves a non-overlapped $95\%$ confidence interval over the single-agent method**. In the task of math word problem generation, there is a slight overlap between ColMAD and Ensemble, for which we obtained a p-value of 6%. It is because both GPT4o-mini and Llama3.1-70B are relatively good at the task; therefore, the performance is relatively saturated.

---

> ### Author Response · Authors · 2025-11-24
> **Response to Reviewer qPyM (part 3)**
>
> > W3.2 F1 scores
>
> **A5** We need to note that **the nature of the error detection task places more importance on the successful identification of errors**, i.e., recall. In fact, there exists a natural trade-off between the accuracy/precision and the recall:
> - For example, a detector can report the existence of errors all the time, which gives 100% recall while low precision (e.g., Llama3.1-8B has 100% recall, but 61% precision on math problem generation);
> - A detector can also report errors when it is very confident, which gives a high precision and low recall (e.g., DeepSeek-R1 has 73% precision, but 54% recall on answerability);
> Given all the considerations, we adopt the F2 score[5], which emphasizes recall, as a comprehensive metric.
>
> > W3.3 Diagonal values of Figure 2
>
> **A6** As stated in the caption of Figure 2, the numbers report the ``Error reductions of prevalent LLMs in detecting errors of GPT-4’’, which is calculated through Eq. (4). We have revised our manuscript to make it clearer. Intuitively, Eq. (4) considers a proxy of an oracle collaboration of two LLMs, which only makes mistakes when both LLMs make mistakes. The numbers of reduced errors are calculated between the oracle proxy and the original single-agent. If the two LLMs are the same, **the oracle proxy will have no difference as the single-agent one, and has 0 reduced errors**.
>
> **References**
>
> [1] Strategic Information Transmission, Econometrica, 1982.
>
> [2] Concrete Problems in AI Safety, arXiv’16.
>
> [3] When Can LLMs Actually Correct Their Own Mistakes? A Critical Survey of Self-Correction of LLMs, TACL’24.
>
> [4] Evaluating LLMs at Detecting Errors in LLM Responses, COLM’24.
>
> [5] Modern Information Retrieval: The concepts and technology behind search, 2011.

---

> ### Author Response · Authors · 2025-11-27
> **A gentle reminder**
>
> Dear Reviewer qPyM,
>
> Thank you once again for your time and effort in reviewing our work. As the discussion period is only one week away, to allow us sufficient time to address your concerns, could you kindly review our responses and let us know if you still have any questions? Thank you!
>
> Best regards,
> Authors

---

### Author Response · Authors · 2025-12-02
**Summary of Rebuttal**

Dear Reviewers and Area Chairs,

We thank the reviewers for their efforts and constructive comments in reviewing our work. We also sincerely appreciate the area chairs for the high workload due to the OpenReview bug.
Although the reviewers didn’t get the chance to reply to us before the bug period, to facilitate the decision, we provide a summary of our main contributions and responses below.

---
## Contributions of this work

The main contributions of this work lie in
- We are the first to investigate the use of MAD for the task of error detection.
- We develop the first theoretical formulation of MAD for error detection, identify the drawbacks of competitive MAD (CopMAD) due to its zero-sum nature, and demonstrate the existence of ``debate hacking’’.
- We then propose a new MAD protocol termed collaborative MAD (ColMAD) to mitigate the issue of CopMAD, and demonstrate both theoretical advantage as well as significant empirical improvements with extensive experiments.

All reviewers agree with the importance of the problem. We are also encouraged to find that the identification of ``debate hacking’’ is appreciated by Reviewers nyiP, the experiments and empirical improvements are widely acknowledged by Reviewers nyiP, 3NPC, and Yicp.

---
## Questions and our responses

While initially this work received one positive recommendation and three negative recommendations, we believe there are some factual misunderstandings and the major concerns shared across reviewers are addressable.

---
### Relations with previous MAD works (qPyM,3NPC,Yicp)

Essentially, previous MAD approaches can be categorized as competitive MAD (CopMAD) [1,2,3,4] and consensus-seeking MAD (CosMAD) [5,6]. A full table of the related work can be found in [the response to Reviewer 3NPC (part 1)](https://openreview.net/forum?id=W6qSjvTQMW&noteId=Vmg3OmjWGp).

Given the categorization, we show that **ColMAD has provable theoretical advantages over CopMAD and CosMAD**. Full theoretical results can be found in [the response to Reviewer 3NPC (part 2)](https://openreview.net/forum?id=W6qSjvTQMW&noteId=yBcWjCQZcK).

In addition, we also extended our baselines to include more previous MAD works following the suggestions by Reviewers 3NPC and Yicp. The results can be found in the [the response to Reviewer 3NPC (part 3)](https://openreview.net/forum?id=W6qSjvTQMW&noteId=Ae3MpbvWUs), where ColMAD demonstrates the **consistent advantages across all baseline methods**.

---
### Relations with previous scalable oversight works (qPyM)

This is a critical factual misunderstanding. Throughout the work, we focus on **improving error detection to achieve scalable oversight**, which follows the spirit of the scalable oversight literature to provide feedback to powerful AI systems[7].

Nevertheless, following the suggestion by Reviewer qPyM, we also extended experiments to MAD protocols with equally powerful debaters, as well as with weaker judges. The results are aligned with our discussion that the heterogeneity of debaters is crucial for the success of error detection.

---
### 95% confidence intervals (qPyM, 3NPC, YiCP)

This is a critical factual oversight. To ensure reproducibility, we set a temperature of 0 for all experiments. Nevertheless, to investigate other settings, we also conducted experiments with GPT4o-mini and Llama3.1-70B with the temperature set to 1 with 10 different runs. The results can be found in the [the response to Reviewer qPyM (part 2)](https://openreview.net/forum?id=W6qSjvTQMW&noteId=VXyJCwNEcY ), It can be found that **ColMAD is the only method that achieves a non-overlapped 95% confidence interval over the single-agent method**.

---
### Fine-grained analysis of debating transcripts (Yicp)

To show that ColMAD produces more human-aligned explanations, we analyze the debating results from CopMAD and ColMAD (Llama3.1-70B vs DeepSeek-R1) in math problem generation. We extract some representative mistakes given by human explanations. The results can be found in [the response to Reviewer Yicp (part 4)](https://openreview.net/forum?id=W6qSjvTQMW&noteId=mcU6paSqU6 ), where we show that **ColMAD consistently identifies the same mistakes pointed out by humans better than other methods**.

---
### Remaining concerns

We also provided point-by-point clarifications and ablation studies to address the remaining concerns. Details can be found in our individual responses to each reviewer.

---
**References**

[1] AI safety via debate, arXiv'18.

[2] Debating with more persuasive LLMs leads to more truthful answers, ICML'24.

[3] On scalable oversight with weak LLMs judging strong LLMs, NeurIPS'24.

[4] Encouraging divergent thinking in large language models through multi-agent debate, EMNLP'24.

[5] Improving factuality and reasoning in language models through multi-agent debate, ICML'24.

[6] Reconcile: Round-table conference improves reasoning via consensus among diverse llms, ACL'24.

[7] Concrete Problems in AI Safety, arXiv’16.

---

### Meta-Review · Area_Chair_6VZf · 2026-01-07

**Summary:**

This paper studies error detection in LLM outputs and argues that standard competitive multi-agent debate can degrade performance via debate hacking, while a collaborative protocol yields more informative critiques and better final judgments. Strength-wise, multiple reviewers agree the problem is important, the framing around debate hacking is useful, and the experimental coverage across models/tasks is broad. The main weaknesses are that the theory is viewed by multiple reviewers as a bit weak, and the novelty is arguably mostly in prompting/framing rather than a new mechanism.

**Reviewer Concerns:**

Concerns about missing some important details were substantially addressed in the rebuttal. Still outstanding are (i) whether the theory adds real insight, (ii) whether the main gains over Ensemble are fundamentally more than extra inference-time compute, and (iii) whether the paper’s positioning vs prior non-competitive/consensus MAD is fully convincing (raised strongly by multiple reviewers, and only partially resolved by added definitions/tables).

**Reviewer Scores:**

I expect some reviewers will increase the scores slightly, but overall still on the negative side.

---

### Decision · Program_Chairs · 2026-01-26

Reject